# EVADING ADVERSARIAL EXAMPLE DETECTION DEFENSES WITH ORTHOGONAL PROJECTED GRADIENT DESCENT

**Oliver Bryniarski**[*]
UC Berkeley

**Nabeel Hingun**[*]
UC Berkeley

**Pedro Pachuca**[*]
UC Berkeley

**Vincent Wang**[*]
UC Berkeley

**Nicholas Carlini**
Google

## ABSTRACT

Evading adversarial example detection defenses requires finding adversarial examples that must simultaneously (a) be misclassified by the model and (b) be detected as non-adversarial. We find that existing attacks that attempt to satisfy multiple simultaneous constraints often over-optimize against one constraint at the cost of satisfying another. We introduce *Selective Projected Gradient Descent* and *Orthogonal Projected Gradient Descent*, improved attack techniques to generate adversarial examples that avoid this problem by orthogonalizing the gradients when running standard gradient-based attacks. We use our technique to evade four state-of-the-art detection defenses, reducing their accuracy to 0% while maintaining a 0% detection rate.

## 1 INTRODUCTION

Generating *adversarial examples* (SZS+14; BCM+13), inputs designed by an adversary to cause a neural network to behave incorrectly, is straightforward. By performing input-space gradient descent (CW17b; MMS+17), it is possible to maximize the loss of arbitrary test examples. This process is both efficient and highly effective. But despite much effort by the community, attempts at designing defenses against adversarial examples have been largely unsuccessful and gradient-descent attacks continue to circumvent new defenses (ACW18; TCBM20).

As a result, many defenses aim to make generating adversarial examples more difficult by requiring additional constraints on inputs for them to be considered successful. Defenses that rely on *detection*, for example, will reject inputs if a secondary detector model determines the input is adversarial (MGFB17; XEQ17). Turning a benign input $x$ into an adversarial example $x'$ thus now requires fooling both the original classifier, $f$, *and* the detector, $g$, simultaneously.

Traditionally, this is done by constructing a single loss function $\mathcal{L}$ that jointly penalizes the loss on $f$ and the loss on $g$ (CW17a), e.g., by defining $\mathcal{L}(x') = \mathcal{L}(f) + \lambda\mathcal{L}(g)$ and then minimizing $\mathcal{L}(x')$ with gradient descent. Unfortunately, many evaluations using this strategy have had limited success—not only must $\lambda$ be tuned appropriately, but the gradients of $f$ and $g$ must also be well behaved.

**Our contributions.** We develop a new attack technique designed to construct adversarial examples that simultaneously satisfy multiple constraints. Our attack approach is a modification of standard gradient descent (MMS+17) and requires changing just a few lines of code. Given two objective functions $f$ and $g$, instead of taking gradient descent steps that optimize the joint loss function $f + \lambda g$, we selectively take gradient descent steps on either $f$ or $g$. This makes our attack both simpler and easier to analyze than prior attack approaches.

We use our technique to evade four state-of-the-art and previously-unbroken defenses to adversarial examples: the Honeypot defense (CCS'20) (SWW+20), Dense Layer Analysis (IEEE Euro S&P'20) (SKCB19), Sensitivity Inconsistency Detector (AAAI'21) (TZLD21), and the SPAM detector presented in Detection by Steganalysis (CVPR'19) (LZZ+19). In all cases, we successfully reduce the accuracy of the protected classifier to 0% while maintaining a detection AUC of less than 0.5—meaning the detector performs worse than random guessing.

---

[*]Equal contributions. Authored alphabetically.

## 2 BACKGROUND

### 2.1 NOTATION

We consider classification neural networks $f : \mathbb{R}^d \to \mathbb{R}^n$ that receive a $d$-dimensional input vector (in this paper, images) $x \in \mathbb{R}^d$ and output an $n$-dimensional prediction vector $f(x) \in \mathbb{R}^n$. We let $g : \mathbb{R}^d \to \mathbb{R}$ denote some other function which also must be considered, where $g(x) \leq 0$ when the constraint is satisfied and $g(x) > 0$ if it is violated. Without loss of generality, in a detection defense this function $g$ is the detector and higher values corresponding to higher likelihood of the input being an adversarial example. To denote the true label of $x$ is given by $y$ we write $c(x) = y$. In an abuse of notation, write $y = f(x)$ to denote the arg-max most likely label under the model $f$.

### 2.2 ADVERSARIAL EXAMPLES

Adversarial examples (SZS[+]14; BCM[+]13) have been demonstrated in nearly every domain in which neural networks are used. (ASE[+]18; CW18; HPG[+]17) Given an input $x$ corresponding to label $c(x)$ and classifier $f$, an adversarial example is a perturbation $x'$ of the input such that $d(x, x') < \epsilon$ and $f(x') \neq c(x)$ for some metric $d$. Additionally, an adversarial example can be targeted if, given a target label $t \neq c(x)$ we have $f(x') = t$ with $d(x, x') < \epsilon$. The metric $d$ is most often that induced by a $p$-norm, typically either $|| \cdot ||_2$ or $|| \cdot ||_\infty$. With small enough perturbations under these metrics, the adversarial example $x'$ is not perceptibly different from the original input $x$.

**Datasets.** We attack each defense on the dataset that it performs best on. All of our defenses operate on images. For three of these defenses, this is the CIFAR-10 dataset (KH09), and for one, it is the ImageNet dataset (DDS[+]09). For each defense we attack, we constrain our adversarial examples to the threat model originally considered to perform a fair re-evaluation, but also generate adversarial examples with standard norms used extensively in prior work in order to make cross-defense evaluations meaningful. We perform all evaluations on a single GPU. Our attacks on CIFAR-10 require just a few minutes, and for ImageNet a few hours (primarily due to the defense having a throughput of one image per second).

### 2.3 DETECTION DEFENSES

We focus our study on detection defenses. Rather than directly improve the robustness of the model (MMS[+]17; RSL18; LAG[+]19; CRK19), detection defenses classify inputs as adversarial or benign (MGFB17; XEQ17) so they can be rejected. While there have been several different strategies attempted to detect adversarial examples in the past (GSS15; MGFB17; FCSG17; XEQ17; MC17; MLW[+]18; RKH19), many of these approaches were broken with adaptive attacks that designed new loss functions tailored to each defense (CW17a; TCBM20).

### 2.4 GENERATING ADVERSARIAL EXAMPLES WITH PROJECTED GRADIENT DESCENT

Projected Gradient Descent (MMS[+]17) is a powerful first-order method for finding such adversarial examples. Given a loss $\mathcal{L}(f, x, t)$ that takes a classifier, input, and desired target label, we optimize over the constraint set $S_\epsilon = \{z : d(x, z) < \epsilon\}$ and solve

$$x' = \arg \min_{z \in S_\epsilon} \mathcal{L}(f, z, t) \tag{1}$$

by taking steps $x_{i+1} = \Pi_{S_\epsilon}(x_i - \alpha \nabla_{x_i} \mathcal{L}(f, x_i, t))$. Here $\Pi_{S_\epsilon}$ denotes projection onto the set $S_\epsilon$, $\alpha$ is the step size, and $x_0$ is randomly initialized (MMS[+]17). This paper adapts PGD in order to solve optimization problems which involve minimizing multiple objective functions simultaneously. For notational simplicity, in the remainder of this paper we will omit the projection operator $\Pi_{S_\epsilon}$.

**Attacks using PGD.** Recent work has shown that it is possible to attack models with *adaptive attacks* that target specific aspects of defenses. For detection defenses this process is often *ad hoc*, involving alterations specific to each given defense (TCBM20). An independent line of work develops automated attack techniques that are reliable indicators of robustness (CH20); however, in general, these attack approaches are difficult to apply to detection defenses.

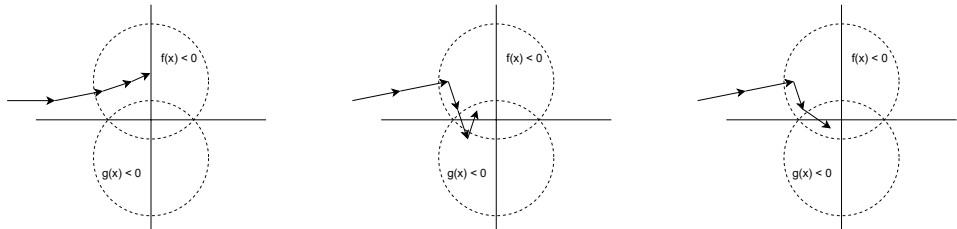

Figure 1: A visualization showing how a standard Lagrangian attack fails when ours succeeds over a non-convex loss landscape. Given two circular regions corresponding to when $f(x) < 0$ (above) and $g(x) < 0$ (below), we would like to find the central region where both are satisfied. **(left)** With Lagrangian PGD, the attack falls in a local minimum and fails to satisfy both constraints simultaneously regardless of the value $\lambda$ selected. **(middle)** Our S-PGD attack first moves towards the upper region by minimizing $f(x)$. Once this constraint is satisfied (and $f(x) < 0$), we begin to minimize $g(x)$; however this overshoots to a point where $f(x) > 0$. A final step recovers a valid solution to both. **(right)** Our O-PGD attack follows the same trajectory for the first two steps optimizing $f(x)$. However after this it takes steps orthogonal to $f(x)$ maintaining the constraint $f(x) < 0$ while simultaneously minimizing $g(x)$, giving a valid solution more quickly.

## 3 RETHINKING ADVERSARIAL EXAMPLE DETECTION

Before we develop our improved attack technique to break adversarial example detectors, it will be useful to understand why evaluating adversarial example detectors is more difficult than evaluating standard classifiers.

Early work on adversarial examples often set up the problem slightly differently than we do above in Equation 1. The initial formulation of an adversarial example (SZS$^+$14; CW17b) asks for the smallest perturbation $\delta$ such that $f(x + \delta)$ is misclassified. In the targeted case, this means solving

$$\arg \min \|\delta\|_2 \text{ such that } f(x + \delta) = t$$

for a given target $t$ that is not equal to the original correct label for $x$. Solving this problem as stated is intractable. It requires searching over a nonlinear constraint set, which is not feasible for standard gradient descent. As a result, detection papers typically (SWW$^+$20; SKCB19) reformulate the search with a Lagrangian relaxation

$$\arg \min \|\delta\|_2 + \lambda \mathcal{L}(f, x + \delta, t) \tag{2}$$

This formulation is simpler, but still (a) requires tuning $\lambda$ to work well, and (b) is only guaranteed to be correct for convex functions $\mathcal{L}$—that it works for non-convex models like deep neural networks is not theoretically justified. It additionally requires carefully constructing loss functions $\mathcal{L}$ (CW17b).

Equation 1 simplifies the setup considerably by exchanging the constraint and objective. Whereas in Equation 2 we search for the smallest perturbation that results in misclassification, Equation 1 instead finds an input $x + \delta$ with $\delta < \epsilon$ that minimizes the classifier's loss. This is a simpler formulation because now the constraint is convex, and so we can run standard gradient descent optimization.

**Evading detection defenses is difficult because there are now two non-linear constraints**. Not only must the input be constrained by a distortion bound and be misclassified by the base classifier, but we must *also* have that they are not detected, i.e., with $g(x) < 0$. This new requirement is nonlinear, and now it becomes impossible to side-step the problem by merely swapping the objective and the constraint as we did before: there will always be at least one constraint that is a non-linear function, and so standard gradient descent techniques can not directly apply.

In order to resolve this difficulty, the existing literature applies the same Lagrangian relaxation as was previously applied to constructing minimum-distortion adversarial examples. That is, breaking a detection scheme involves solving

$$\arg \min_{x \in S_\epsilon} \mathcal{L}(f, x, t) + \lambda g(x) \tag{3}$$

where $\lambda$ is a hyperparameter that controls the relative importance of fooling the classifier versus fooling the detector. However, this formulation again brings back all of the reasons why the community moved past minimum-distortion adversarial examples.

## 3.1 A MOTIVATING EXAMPLE

Let $f(\vec{x}) = \exp(-1) - \exp(-\|\vec{x} - \vec{e}\|_2^2) - \varepsilon$ and $g(\vec{x}) = \exp(-1) - \exp(-\|\vec{x} + \vec{e}\|_2^2) - \varepsilon$ where $\vec{e} \in \mathbb{R}^N$ and $\|\vec{e}\| = 1$, as visualized in Figure 1. By setting $\varepsilon$ to a small constant, the only solution that satisfies both $f(\vec{x}) < 0$ and $g(\vec{x}) < 0$ can be made arbitrarily close to the origin $\vec{x} = \vec{0}$.

However, no standard Lagrangian formulation will be able to find this solution. Consider the sum $h(\vec{x}; \lambda) = f(\vec{x}) + \lambda g(\vec{x})$; then we can show that for all $\lambda$ we will have $\arg\min_{\vec{x}} h(\vec{x}; \lambda) \neq 0$. To see this, observe that while it is possible for the gradient $\nabla h(\vec{0}; \lambda) = 0$ (one of the conditions for a value to be a local minima), the loss surface is always "concave down" at the origin. It will always be possible to move slightly closer to $\vec{e}$ or $-\vec{e}$ and decrease the loss. Therefore, minimizing $h(x)$ will never be able to find a valid stable solution to this extremely simple problem, as it will always collapse to finding a solution of either $\vec{e}$ or $-\vec{e}$, which only satisfies one of the two equations.

## 4 OUR ATTACK APPROACHES

We now present our attack strategy designed to generate adversarial examples that satisfy two constraints. As we have been doing, each of our attack strategies defined below generates a targeted adversarial example $x'$ so that $f(x') = t$ but $g(x') < 0$. Constructing an untargeted attack is nearly identical except for the substitution of maximization instead of minimization.

### 4.1 SELECTIVE GRADIENT DESCENT

Instead of minimizing the weighted sum of $f$ and $g$, our first attack never optimizes against a constraint once it becomes satisfied. That is, we write our attack as

$$\mathcal{A}(x, t) = \arg\min_{x': \|x - x'\| < \epsilon} \underbrace{\mathcal{L}(f, x', t) \cdot \mathbb{1}[f(x) \neq t] + g(x') \cdot \mathbb{1}[f(x) = t]}_{\mathcal{L}_{\text{update}}(x, t)}. \tag{4}$$

The idea here is that instead of minimizing a convex combination of the two loss functions, we selectively optimize either $f$ or $g$ depending on if $f(x) = t$, ensuring that updates are always helping to improve either the loss on $f$ or the loss on $g$.

Another benefit of this style is that it decomposes the gradient step into two updates, which prevents *imbalanced gradients*, where the gradients for two loss functions are not of the same magnitude and result in unstable optimization (JMW[+]20). In fact, our loss function can be viewed directly in this lens as following the margin decomposition proposal (JMW[+]20) by observing that

$$\nabla \mathcal{L}_{\text{update}}(x, t) = \begin{cases} \nabla \mathcal{L}(f, x, t) & \text{if } f(x) \neq t \\ \nabla g(x) & \text{if } f(x) = t. \end{cases} \tag{5}$$

That is, each iteration either take gradients on $f$ or on $g$ depending on whether $f(x) = t$ or not.

Recalling the motivating example from Figure 1, this selective optimization formulation would be able to find a valid solution. No matter where we initialized our adversarial example search, minimizing with respect to $f(x)$ will eventually give a valid solution near $e$. Once this happens, we then switch to optimizing against $g(x)$ (because $f(x)$ is satisfied). From here we will eventually converge on the solution $x \approx \vec{0}$.

### 4.2 ORTHOGONAL GRADIENT DESCENT

The prior attack, while mathematically correct, might encounter numerical stability difficulties. Often, the gradients of $f$ and $g$ point in opposite directions, that is, $\nabla f \approx -\nabla g$. As a result, every step spent optimizing $f$ causes backwards progress on optimizing against $g$. This results in the optimizer constantly "undoing" its own progress after each step that is taken. To address this problem, we would like to "remove" the portion of one gradient optimization step that "undoes" the progress of a previous optimization step.

To do this, we call on vector projections. Note that $\nabla g(x)^{\perp} = \nabla \mathcal{L}(f, x, t) - \text{proj}_{\nabla g(x)} \nabla \mathcal{L}(f, x, t)$ is orthogonal to the gradient $\nabla g(x)$, and similarly $\nabla \mathcal{L}(f, x, t)^{\perp}$ is orthogonal to $\nabla \mathcal{L}(f, x, t)$. We

integrate this fact with Equation 5 to give a slightly different update rule that again solves Equation 4, however this time by optimizing:

$$\mathcal{L}_{\text{update}}(x,t) = \begin{cases} \nabla \mathcal{L}(f,x,t) - \text{proj}_{\nabla g(x)} \nabla \mathcal{L}(f,x,t) & \text{if } f(x) \neq t \\ \nabla g(x) - \text{proj}_{\nabla \mathcal{L}(f,x,t)} \nabla g(x) & \text{if } f(x) = t. \end{cases} \tag{6}$$

The purpose of this update is to take gradient descent steps with respect to one of $f$ or $g$ in such a way that we do not significantly disturb the loss of the function not chosen. In this way, we prevent our attack from taking steps that undo work done in previous iterations of the attack.

It is also important to note that, in the high-dimensional space in which a typical neural network operates, the gradients of $f$ and $g$ are practically never exactly opposite, that is a situation where $\nabla f = -\nabla g$. In this case, the projection of $\nabla f$ onto $\nabla g$ and $\nabla g$ onto $\nabla f$ would be 0 and we could not make any meaningful optimizations towards satisfying either constraint with OPGD.

Again recalling Figure 1, by taking steps that are orthogonal to $f(x)$ we can ensure that once we reach the acceptable region for $f$, we never leave it, and much more quickly converge on an adversarial example that evades detection.

## 5 CASE STUDIES

We validate the efficacy of our attack by using it to circumvent four previously unbroken, state-of-the-art defenses accepted at top computer security or machine learning venues. Three of the case studies utilizes models and code obtained directly from their respective authors. In the final case the original authors provided us with matlab source code that was not easily used, which we re-implemented.

**Attack Success Rate Definition.** We evaluate the success of our attack by a standard metric called *attack success rate at N* (SR@N for short) (SKCB19). We use SR@N to ensure comparability across different case studies but more importantly between our results and our case studies' original results. SR@N is defined as the fraction of *targeted attacks* that succeed when the defense's false positive rate is set to $N\%$. (To adjust a defense's false positive rate it suffices to adjust the detection threshold $\phi$ so that inputs are rejected as adversarial when $g(x) > \phi$.) For example, a $94\%$ $SR@5$ could either be achieved through $94\%$ of inputs being misclassified as the target class and $0\%$ being detected as adversarial, or by $100\%$ of inputs being misclassified as the target class and $6\%$ being detected as adversarial, or some combination thereof. We report SR@5 and SR@50 for our main results [1], and for completeness also give the full ROC curve of the detection rate for a more complete analysis.

**Attack Hyperparameters.** We use the same hyperparmaeter setting for all attacks shown below. We set the distortion bound $\varepsilon$ to 0.01 and .031; several of these papers exclusively make claims using the value of 0.01 (SWW[+]20; SKC[+]20; TZLD21), but the value $0.031 = 8/255$ is more typical in the literature (TCBM20). We run our attack for $N = 1000$ iterations of gradient descent with a step size $\alpha = \frac{\varepsilon}{10}$ (that is, the step size changes as a function of $\varepsilon$ which follows standard advice (MMS[+]17)).

### 5.1 HONEYPOT DEFENSE

The first paper we consider is the Honeypot Defense (SWW[+]20). Instead of preventing attackers from directly constructing adversarial examples, the authors propose to *lure* attackers into producing specific perturbations that are easy to find and hard to ignore. These perturbations are called "honeypots" or trapdoors and can be easily identified by a detector. For their evaluation on the MNIST and CIFAR-10 dataset, the authors use 5 sets of randomly selected $3 \times 3$ squares per label.

Formally, consider an input $x$ to the classifier, $f$. During training, $f$ is injected with a honeypot, $\Delta$. The signature of a particular honeypot, $S_\Delta$, is the expectation of the neuron activations of $f$ over multiple sample inputs containing $\Delta$. During inference, the internal neuron activation pattern $e(x)$ is compared to $S_\Delta$ using cosine similarity. Specifically, for a predefined threshold $\phi$, if $\cos(e(x), S_\Delta) > \phi$, then $x$ is flagged as adversarial. One additional modification the authors make is to use neuron randomization. When creating the signatures, a random sample of neuron activations

---

[1]The value $5\%$ is used in many prior defenses in the literature (MLW[+]18; XEQ17), and $50\%$ is an extreme upper bound and would reduce the model's accuracy by half.

| Attack | eps=0.01 | | eps=0.031 | |
|---|---|---|---|---|
| | SR@5 | SR@50 | SR@5 | SR@50 |
| (SWW$^+$20) | 0.02 | - | - | - |
| Reproduction | 0.13* | - | - | - |
| Orthogonal | **1.0** | **0.93** | **1.0** | **0.92** |
| Selective | 0.998 | 0.505 | 0.996 | 0.628 |

(a) Attack success rate for our two proposed attacks. *When we reproduce the authors attack, we reach slightly a higher attack success rate of 13%.

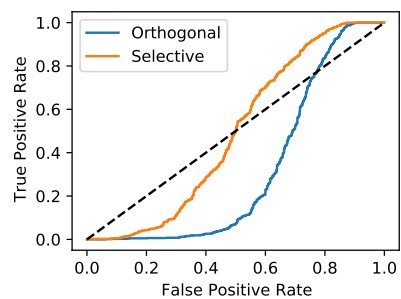

(b) Defense ROC, $\varepsilon = 0.01$, AUC of 0.30

Figure 2: Honeypot attack evaluation. Compared to the originally reported 2% success rate, our attack reaches a 100% attack success rate under the same distortion bound. While the ROC curve does cross over the $x = y$ line, this only occurs after a FPR of 70% which is completely unusable.

is ignored and the same set of neurons is ignored when calculating the cosine similarity during adversarial attack detection.

We test our method of generating adversarial images on CIFAR-10 with an $l_\infty$ bound of $\epsilon = \frac{8}{255}$. The original model has a baseline accuracy of .87. To test the defense, we generate targeted attacks choosing the target from among the incorrect labels. As a simple baseline attack, we confirm that FGSM achieves an accuracy of .06 on the classifier but the ROC AUC of the detector is .994.

To evaluate the efficacy of our attack, we use the authors pre-trained code, and convert their models from TensorFlow to PyTorch to run our attacks. We find that our attack is extremely effective and has an attack success rate above 99.9%, even at a 50% false positive rate, and an AUC of 0.30. In Table 2 we give the attack results for several configurations of our attack. We shared our results with the original defense authors who confirmed our adversarial examples successfully evaded their detection (an important step, given that we had converted the model to PyTorch).

## 5.2    DENSE LAYER ANALYSIS

Many recent defenses analyze the hidden activations of neural networks while processing benign and adversarial inputs (e.g., following (MGFB17)). These defenses aim to capitalize on differences in activation patterns among benign and adversarial inputs to train a separate classifier capable of detecting inputs as adversarial.

The most recent paper in this direction Sperl *et al.* extract dense layer activation patterns among benign and adversarial inputs and train a secondary binary classifier that detects adversarial examples (SKCB19). The authors do this by first performing a forward pass through a target neural network with both adversarial and benign inputs to create a mixed-feature dataset of activation-label pairs. Then, using the mixed-feature dataset, they train a secondary binary classifier capable of discerning between adversarial and benign inputs. When evaluating their models, the authors pass an input through the target model to obtain the activation feature vectors for a particular input as well as a potential classification. They then pass this feature vector through the secondary classifier. If the secondary classifier alerts that the input was adversarial, the classification is thrown away. Otherwise, classification proceeds as normal.

Sperl *et al.* evaluate this defense with 5 leading adversarial attacks on the MNIST and CIFAR-10 datasets using several models and report high accuracies for benign inputs and high detection rates for adversarial inputs. The authors report a worst-case individual attack accuracy of 0.739.

In accordance with our framework, we assign the cross entropy loss of the classifier to our primary function and binary cross entropy loss of the detector as our secondary function.

We obtain source code and pre-trained defense models from the authors in order to ensure that our attack matches the defense as closely as possible. We now detail the results of our attack at $\epsilon = .01$

| Attack | eps=0.01 | | eps=0.031 | |
| --- | --- | --- | --- | --- |
| | SR@5 | SR@50 | SR@5 | SR@50 |
| (SKC⁺20) | $\leq 0.13^{*}$ | - | - | - |
| Reproduction | $0.20^{+}$ | - | - | - |
| Orthogonal | 0.374 | 0.163 | **1.0** | 0.718 |
| Selective | **0.83** | **0.441** | **1.0** | **0.865** |

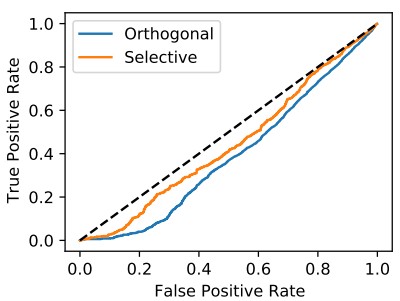

(a) Attack success rate for our two proposed attacks. *The original paper reported only at a 20% FPR, we take this as an upper bound for what could be achieved at 5% FPR. ⁺When we reproduce the authors attack, we reach slightly higher success rate of 20%.

(b) Defense ROC, $\varepsilon = 0.01$, AUC of 0.38

Figure 3: DLA attack evaluation. Our attack succeeds with $83\%$ probability compared to the original evaluation of $13\%$ (with $\varepsilon = 0.01$), and $100\%$ of the time under the more typical $8/255$ constraint.

and at $\epsilon = .03$ at false positive rates of 5% and 50% in Figure 3. We find that our attack is extremely effective, resulting in an accuracy of 0 at a detection rate of 0 with a false positive rate of 5% under $\epsilon = .03$ bounds and an AUC of 0.38. Finally, to validate that our attack succeeded, we again shared the resulting adversarial examples with the authors who confirmed our attack results.

### 5.3 SENSITIVITY INCONSISTENCY OF SPATIAL-TRANSFORM DOMAIN

We next evaluated our attack on the Sensitivity Inconsistency Detector (SID) proposed by Tian *et al.* (TZLD21). This defense relies on the observations of Fawzi *et al.* (FMDFS18) that adversarial examples are movements, in the form of perturbations, of benign inputs in a decision space along an adversarial direction. Tian *et al.* then conjecture that, because adversarial examples are likely to lie near highly-curved decision boundaries, and benign inputs lie away from such boundaries, fluctuations in said boundaries will often result in a change in classification of adversarial examples but not in classification of benign inputs.

To measure sensitivity against decision boundary transformations, Tian *et al.* design a dual classifier which is the composition of a weighted additive wavelet transform layer and a DNN classifier with the same structure as the original classifier. When doing a forward pass of the system, the authors run an input through both the primal and the dual model, then pass both results to the detector that discriminates among adversarial and benign classes. With these models, the authors then define their so-called feature of sensitivity inconsistency $S(x_0)$.

$$S(x_0) = \{f_i(x_0) - g_i(x_0)\}_{i=1}^{K}$$

where $f_i(x_0)$ and $g_i(x_0)$ are the predictions of the primal and the dual respectively. Input $x_0$ is classified as adversarial if $S(x_0)$ is greater than a threshold $\phi$. SID achieves improved adversarial example detection performance, especially in cases with small perturbations in inputs. The authors report a worst-case, individual attack detection AUC % of 0.95.

Now, we want to create adversarial examples that are misclassified by the original model and not flagged as adversarial by the Sensitivity Inconsistency Detector. We assign the loss of our target model to our primary function and the loss of the Sensitivity Inconsistency Detector as our secondary function. The initial target model had an accuracy of .94 and deemed .06 of all inputs adversarial.

We again obtain source code from the authors along with pre-trained models to ensure evaluation correctness. We describe our attack's results at $\epsilon = .01$ and at $\epsilon = .03$ at false positive rates of 5% and 50% in Figure 4. Our attack works well in this case and induces an accuracy of 0 at a detection rate of 0 with a false positive rate of 5% under $\epsilon = .03$ bounds with an AUC of 0.25.

### 5.4 DETECTION THROUGH STEGANALYSIS

Since adversarial perturbations alter the dependence between pixels in an image, Liu *et al.* (LZZ⁺19) propose a defense which uses a *steganalysis*-inspired approach to detect "hidden features" within an

| Attack | eps=0.01 | | eps=0.031 | |
|---|---|---|---|---|
| | SR@5 | SR@50 | SR@5 | SR@50 |
| (TZLD21) | $\leq 0.09^{*}$ | - | - | - |
| Orthogonal | **0.931** | **0.766** | **1.0** | **0.984** |
| Selective | 0.911 | 0.491 | **1.0** | 0.886 |

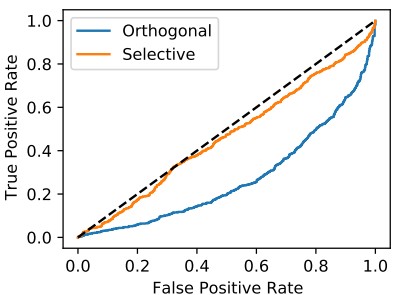

(a) Attack success rate for our two proposed attacks. *The original paper only reports AUC values and does not report true positive/false positive rates. The value of $9\%$ was obtained by running PGD on the author's defense implementation.

(b) Defense ROC, $\varepsilon = 0.01$, AUC of 0.25

Figure 4: SID attack evaluation. Our attack succeeds with $93\%$ probability compared to the original evaluation of $9\%$ under a $\varepsilon = 0.01$-norm constraint, and $100\%$ under a $\varepsilon = 0.031$.

image. These features are then used to train binary classifiers to detect the perturbations. Unlike the prior defenses, this paper evaluates on ImageNet, reasoning that small images such as those from CIFAR-10 and MNIST do not provide enough inter-pixel dependency samples to construct efficient features for adversarial detection, so we attack this defense on ImageNet.

As a baseline, the authors use two feature extraction methods: SPAM and Spatial Rich Model. For each pixel $X_{i,j}$ of an image $X$, SPAM takes the difference between adjacent pixels along 8 directions. For the rightward direction, a difference matrix $A^{\rightarrow}$ is computed so that $A^{\rightarrow}_{i,j} = X_{i,j} - X_{i,j+1}$. A transition probability matrix $M^{\rightarrow}$ between pairs of differences can then be computed with

$$M^{\rightarrow}_{x,y} = Pr(A^{\rightarrow}_{i,j+1} = x | A^{\rightarrow}_{i,j} = y)$$

where $x, y \in \{-T, ..., T\}$, with $T$ being a parameter used to control the dimensionality of the final feature set $F$. We use $T = 3$ in accordance with that used by the authors. The features themselves are calculated by concatenating the average of the non-diagonal matrices with the average of the diagonal matrices:

$$F_{1,...,k} = \frac{M^{\rightarrow} + M^{\leftarrow} + M^{\uparrow} + M^{\downarrow}}{4} \qquad F_{k+1,...,2k} = \frac{M^{\nearrow} + M^{\nwarrow} + M^{\searrow} + M^{\swarrow}}{4}$$

In order to use the same attack implementation across all defenses, we reimplemented this defense in PyTorch (the authors implementation was in matlab). Instead of re-implementing the full Fisher Linear Discriminant (FLD) ensemble (KFH12) used by the authors, we train a 3-layer fully connected neural network on SPAM features and use this as the detector. This allows us to directly investigate the claim that SPAM features can be reliably used to detect adversarial examples, as FLD is a highly non-differentiable operation and is not a fundamental component of the defense proposal.

The paper also proposes a second feature extraction method named "Spatial Rich Model" (SRM) that we do not evaluate against. This scheme follows the same fundamental principle as SPAM in modeling inter-pixel dependencies—there is only a marginal benefit from using these more complex models, and so we analyze the simplest variant of the scheme.

Notice that SPAM requires the difference matrices $A$ to be discretized in order for the dimensionality of the transition probability matrices $M$ to be finite. To make this discretization step differentiable and compatible with our attacks, we define a count matrix $X$ where, for example, $X^{\rightarrow}_{x,y}$ counts, for every pair $i, j$, the number of occurrences of $y$ in $A^{\rightarrow}_{i,j}$ and $x$ in $A^{\rightarrow}_{i,j+1}$. $M^{\rightarrow}_{x,y}$ is then defined by:

$$M^{\rightarrow}_{x,y} = P(A^{\rightarrow}_{i,j+1} = x | A^{\rightarrow}_{i,j} = y) = \frac{X^{\rightarrow}_{x,y}}{\sum_{x'} X^{\rightarrow}_{x',y}}$$

To construct a differentiable approximation, consider without loss of generality the rightward difference matrix $A^{\rightarrow}_1$ for an image. We construct a shifted copy of it $A^{\rightarrow}_2$ so that $A^{\rightarrow}_{2_{i,j}} = A^{\rightarrow}_{1_{i,j+1}}$. We then define a mask $K$ so that

$$K_{i,j} = \mathbb{1}[x \leq A^{\rightarrow}_{2_{i,j}} < x + 1 \cap y \leq A^{\rightarrow}_{1_{i,j}} < y + 1]$$

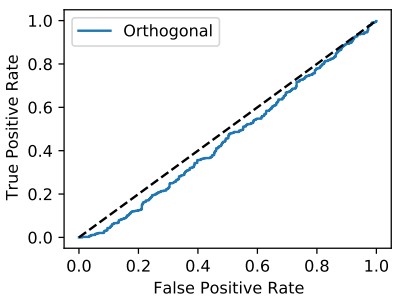

| Attack | eps=0.01 | | eps=0.031 | |
|---|---|---|---|---|
| | SR@5 | SR@50 | SR@5 | SR@50 |
| (LZZ$^+$19) | 0.03 | - | .03 | - |
| Orthogonal | **0.988** | **0.54** | **1.0** | **0.62** |

(a) Attack success rate for our proposed attack. For computational efficiency, we only run our Orthogonal attack as the detection model has a throughput of one image per second.

(b) Defense ROC, $\varepsilon = 0.01$, AUC of 0.44

Figure 5: Steganalysis attack evaluation. We find it difficult to decrease the detection score lower than the original score on the non-adversarial input, thus the AUC is almost exactly 0.5.

Each element of the intermediate matrix $X_{x,y}^{\rightarrow}$ counts the number of pairs in $A_1^{\rightarrow}$ and $A_2^{\rightarrow}$ which would be rounded to $x$ and $y$ respectively after discretization:

$$X_{x,y}^{\rightarrow} = \frac{\sum_{i,j} \left(K \circ A_2^{\rightarrow}\right)_{i,j}}{x}$$

where $\circ$ is the Hadamard product. If we normalize $X^{\rightarrow}$ so that the sum of elements in each column is equal to 1, we get the probability of difference values $x \in A_2^{\rightarrow}$ conditioned on column $y \in A_1^{\rightarrow}$. Thus, for any pair of indices $i, j$,

$$M_{x,y}^{\rightarrow} = P(A_{2_{i,j}}^{\rightarrow} = x | A_{1_{i,j}}^{\rightarrow} = y) = \frac{X_{x,y}^{\rightarrow}}{\sum_{x'} X_{x',y}^{\rightarrow}}$$

Using this differentiable formulation of SPAM feature extraction, we train an auxillary detector as described above and use its gradients to apply our attack on the original, non-differentiable detector.

The authors evaluate their defense on 4 adversarial attacks and report high accuracy for benign inputs and high detection rates for adversarial inputs. The best attack they develop still has a success rate less than 3%. In contrast, our attack on SPAM using the differentiable approximation has a success rate of 98.8% when considering a 5% false positive rate, with an AUC of 0.44, again less than the random guessing threshold of 0.5.

## 6 CONCLUSION

Generating adversarial examples that satisfy multiple constraints simultaneously (e.g., requiring that an input is both misclassified and deemed non-adversarial) requires more care than generating adversarial examples that satisfy only one constraint (e.g., requiring only that an input is misclassified). We find that prior attacks unnecessarily over-optimizes one constraint when another constraint has not yet been satisfied. Our new attack methology is designed to avoid this weakness, and as a result can reduce the accuracy of four previously-unbroken detection methods to 0% accuracy while maintaining a 0% detection rate at 5% false positive rates.

We believe our attack approach is generally useful, but it is not a substitute for trying other attack techniques. We do not envision this attack as a complete replacement for standard Lagrangian-based attacks, but rather a complement; defenses must carefully consider their robustness to both prior attacks as well as this new one. Notice, for example, that for one of the four defenses we study Selective PGD performs better than Orthogonal PGD—indicating these attacks are complementary to each other. Additionally, automated attack tools (CH20) would benefit from adding our optimization trick to their collection of known techniques that could (optionally) compose with other attacks. We discourage future work from blindly applying this attack without properly understanding its design criteria. While this attack is effective for the defenses we consider, it is not the only way to do so, and may not be the correct way to do so in future defense evaluations. Evaluating adversarial example defenses will necessarily require adapting any attack strategies to the defense's design.

## ACKNOWLEDGEMENTS

We thank the authors of the papers we use in the case studies, who helped us answer questions specific to their respective defenses and agreed to share their code with us. We are also grateful to Alex Kurakin and the anonymous reviewers for comments on drafts of this paper.

## ETHICS STATEMENT

All work that improves adversarial attacks has potential negative societal impacts. Yet, we believe that it is better for those vulnerabilities to be known rather than relying on security through obscurity. We have attacked no deployed system, and so cause no direct harm; and by describing how our attack works, future defenses will be stronger. We have communicated the results of our attack to the authors of the papers we break as a form of responsible disclosure, and also to ensure the correctness of our results.

## REPRODUCIBILITY STATEMENT

All of the code we used to generate our results will be made open source in a GitHub repository. The datasets we use (MNIST, CIFAR10, ImageNet) are available online and widely studied. We obtained original copies of the code associated with 3 of the 4 case studies. We used code either directly from the authors or code released publicly alongside an academic paper. In the case of steganalysis, we reimplemented the paper to the best of our ability. We also provide a Python class constructor so that future work can test or improve our results. Again, we relayed results to the authors of each paper and received confirmation that our adversarial examples were indeed adversarial and not detected by the author's original implementations.

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
