# OpenReview forum: "Evading Adversarial Example Detection Defenses with Orthogonal Projected Gradient Descent"
_ICLR.cc/2022/Conference — ICLR 2022 Poster_

### Official Review · Reviewer_K9MM · 2021-10-19

**Correctness:** 3
**Technical Novelty And Significance:** 2
**Empirical Novelty And Significance:** 2
**Recommendation:** 6
**Confidence:** 3

**Details Of Ethics Concerns:**

The authors addressed ethical issues well.

**Main Review:**

The main proposed approaches are a fairly simple update to existing attack methods, though appear to be quite effective.

The presentation of the paper was generally clear, but with some areas of issue. For example
-	Section 2.4 was a bit confusing. It was not clear (to me) from the preceding discussion that the focus was to be on targeted attacks. For example, in equation (1), the minimisation of the loss suggests that this is targeted (to class t). Yet in section 2.2, t was used as the valid class of the data point x.
-	In Section 3, however, it seems that t is the valid class, as the first equation is targeting attacks which result in distinct classifications from t. Also reference in the text just above  the bold heading refers to equation2 as maximising the loss (equation 2 is currently a minimum) – which it would do if t were the valid class.
-	In Section 3.1, the function f as defined is –epsilon at x=e and positive (for small epsilon) at x=0. I think the 1 should probably be something like exp(-1) so that it evaluates to –epsilon at the origin. As it stands this section does not make sense.
-	It does not seem clear in section 3.1, even if it were corrected, why a Lagrangian formulation would not work. (Note that that is an independent statement of whether gradient descent would work). Also it is said that where the gradient of h vanishes wrt x the loss function has a non-zero slope and so the loss can be further minimised. But if one respects the constraint g(x)<0 this may not be true. In fact, surely the solution to minimising f st g<0 is the point on the y-axis at the apex of g’s circle. At this point, the gradient of f will point upwards, but one cannot follow that without violating the constraint. I am not clear what the authors are arguing exactly.
-	For the methods proposed in 4.1 and 4.2, it is not made made clear exactly how the proposed gradients are to be used. Obviously, presumably in a PGD style attack. (But even the PGD attack in section 2.4 is imprecisely specified as it is not made clear exactly what x_0 is chosen to be nor how may steps are to be performed, etc.)
-	It is stated that the proposed attack is “easier to analyse” than prior approaches. Yet no analysis really has been done.

The experimental results seem to be quite convincing (see comments below also), though it would be good to have seen ROC curves for other (standard) attacks included in the figures for comparison (the only data from other attacks is a simple reproduction of the SN@5 value from the original paper (for usually only one epsilon value). Also, given the defence in 5.4 was reimplemented, it would have been good to see a statement that the original results of the original authors were reproduced, and that those authors had also confirmed the attacks, as was said in at least two of the other cases.

The experiments conducted seem to be fair and thorough, in the sense that the attacks are evaluated against the extant defences with both the threat model originally considered for that defence as well as with common parameter choices. In most cases, the original code of the defence’s authors is used, and also in most cases it seems the authors confirmed their attacks as successful with the defence’s authors.

A side query: I accept that the goal is to not have the gradient of one target conflict with the other (i.e. f and g). But to make the gradient used purely orthogonal to the other is a bit extreme and throws away advantageous cases where the change is in the correct direction for both at the same time. Would it not be possible to instead simply use the orthogonal gradient in cases where its projection is in the negative direction on the other gradient, and use it unchanged otherwise, to get the best of both worlds.

Though the results are good, the issues noted above with regard to clarity and clear motivation mean that my score (for now) is lowered.

Some minor comments:
-	Should “defences” in the third paragraph of the Introduction be “attacks”?
-	In section 3.1, h(x,lamda)=0 uses vector notation on the 0. Isn’t h() a scalar?
-	I found the notation confusing for projections (it may just be me!). But the paper seems to use proj_A B to denote the projection of the vector A along the direction of the vector B. This would seem counterintuitive i.e. A and B wrong way round? Not sure what is standard here, so I could be wrong.
-	In section 4.2, “... exactly opposite, that is a situation where grad f = grad g”. Surely you mean “- grad g”?
-	Section 5: “Three of the case study” should be “Three of the case studies”
-	The text “The equivalence ...” after equation (5) is trivial and need not be said.
-	FLD not defined??
-	In section 5.4, the specification of the two ranges of features is the same. Should some be diagonal?
-	“non-diiferentiable”
-	“any every”
-	Why is only one of the proposed approaches used in 5.4?
-	In the conclusion it is claimed that 0% detection was achieved with 0% accuracy at 5% FPR. This was mentioned in one case, but not clearly achieved in the others? Or at least it wasn’t noted explicitly for the other cases.


**Summary Of The Paper:**

This paper considers the problem of finding adversarial examples that simultaneously defeat a detector of adversarial examples. An argument is made that existing attacks often achieve one goal at the expense of another. This argument motivates the proposal of two attack techniques. These are evaluated against four existing detection-based defence methods, with successful results.

**Summary Of The Review:**

Overall, the two proposed techniques, though a relatively simple update to existing approaches, seem to perform well and provide an effective attack against state of the art adversarial attack detection techniques. However, the presentation of the paper is a bit confused in places and the motivation for the approach a bit unclearly argued. It is hard to argue against such convincing numerical results, so with some rework to address the issues raised I think my recommendation for the paper could increase.

Edit: Score increased after author response and revision.

---

> ### Author Response · Authors · 2021-11-18
> **Author Response**
>
> We thank the reviewer for their comments.
>
> > various clarifications about the notation of targeted vs untargeted attacks
>
> We agree that our presentation of attacks as targeted or untargeted and our usage of *t* as the valid class or target class can be confusing. Accordingly, we have amended sections 2.4 and 3 so that attacks are framed as targeted throughout the paper and *t* is used only as the target class.
>
> > various questions about Section 3.2 and Figure 1
>
> The reviewer is correct that this math is somewhat confusing. We’ll simplify things somewhat differently, by just folding all of the constants “1+\epsilon” into just “\epsilon” and then say “for an appropriate value of \epsilon, the only solution that exists is close to the origin”. It’s easy to see this is true, and hopefully because we’re just trying to illustrate what’s going on it will help clarify without bogging the reader down in extra details that are less important.
>
> > why a Lagrangian formulation would not work
>
> The reviewer is correct that “minimizing f st g<0” would find the valid solution. But it is not actually possible to have a constraint “g<0” in practice, because this is a complex neural network (as we note in the paper). So one thing that could be done is to never move outside of the valid region for f once a solution on the interior is found (and this is exactly what our projected gradient descent idea does!)
>
> However the statement we’re making in the paper is slightly different than what the reviewer is reading. We are saying the *lagrangian* formulation can not work. This formulation minimizes “f + \lambda g”, and we believe that we are correct here that this formulation can not find an adversarial example for our example.
>
> > “how the proposed gradients are to be used”
>
> We leave PGD unchanged except that the gradient step is replaced with our gradient computation. So in particular this means x_0 is a random init as defined by Madry et al. 2017. We have amended section 2.4 to make x_0 explicit. We use 1000 steps as specified in “Attack Hyperparameters” (Section 5).
>
> > It is stated that the proposed attack is “easier to analyse”
>
> By “easier to analyze” we mean that, when the attack fails, it is easier to interpret what is going wrong and make adjustments accordingly. We will clarify this in the paper.
>
> > Would it not be possible to instead simply use the orthogonal gradient in cases where its projection is in the negative direction on the other gradient, and use it unchanged otherwise, to get the best of both worlds.
>
> It would certainly be possible to use the orthogonal gradient only when the gradients oppose each other and otherwise leave the gradient unchanged. This would be an interesting extension that merges together our two attacks; we have not significantly explored this direction because these simpler attacks were sufficient to break the defenses.
>
> > other comments
>
> Thank you for carefully reviewing the paper and your minor comments. We have addressed them in the revision of the paper. As outlined in 5.4, we only evaluate one of the steganalysis-based approaches because the other schemes follow the same principle as SPAM
> in that they extract features by modeling inter-pixel dependencies. Since there is only marginal benefit from analyzing these more complex yet similar models, we only evaluate our attack on SPAM. Our claim of 0% detection rate and 0% accuracy is for eps = 0.03, which is bolded in each table. It's correct that this was not explicitly stated in some cases since in each case we wrote our results in comparison to the epsilon value used by the original authors for fairness.

---

> > ### Comment · Reviewer_K9MM · 2021-11-22
> > **Response**
> >
> > Thanks to the authors for their responses to my questions and comments. As indicated in my summary, I feel I can (and have) increased my score as a consequence.

---

### Official Review · Reviewer_ndnu · 2021-10-22

**Correctness:** 4
**Technical Novelty And Significance:** 3
**Empirical Novelty And Significance:** 3
**Recommendation:** 8
**Confidence:** 4

**Main Review:**

I find this paper pleasant to read in general: SPGD and OPGD are well motivated and clearly explained, while the experimental verification is sufficient and convincing. I only have several minor comments:

1. How effective are SPGD and OPGD against an ensemble of detectors instead of just one?
2. For Honeypot Defence and Sensitivity Inconsistency Detector, what is the impact of the threshold $\phi$ on the effectiveness of SPGD and OPGD?
3. Why is OPGD much less effective than SPGD against Dense Layer Analysis when $\epsilon$=0.01 (Figure 3)?
4. Why isn’t SPGD evaluated against Steganalysis?

**Summary Of The Paper:**

This paper proposes two techniques for generating adversarial examples: Selective Projected Gradient Descent (SPGD) and Orthogonal Projected Gradient Descent (OPGD). In order to fool both the victim model $f$ and a detector $g$, SPGD selectively optimise either $f$ or $g$ depending on whether the modified input is misclassified as the target class, while OPGD further orthogonalizes the gradients. Evaluation on four previously unbroken, state-of-the-art defence methods demonstrate the effectiveness of the proposed attacks.

**Summary Of The Review:**

The proposed attacks for creating adversarial samples are simple but effective, and the evaluation is sufficient.

---

> ### Author Response · Authors · 2021-11-18
> **Author Response**
>
> We thank the reviewers for their helpful clarification questions.
>
> > How effective are SPGD and OPGD against an ensemble of detectors instead of just one?
>
> While writing the paper we had an idea in mind about how to do this, but there aren’t interesting recently-published detection defenses which involve multiple detector models that we are aware of, so we did not highlight this aspect in our text. Here is how we would first try to do so. For selective gradient descent, rather than selecting the movement direction by selecting between the two constraint functions f and g, we can instead select for the constraint function with the highest value (given that we wish to minimize them all) out of a collection \{g_i\}. For orthogonal gradient descent, we can use Gram-Schmidt to project \nabla g_i onto \span{\nabla g_1, \ldots,\nabla g_{i-1}, \nabla g_{i+1}, \ldots, \nabla g_n}, following the intuition that we take steps which preserve the value of g_1, \ldots, g_{i-1},  g_{i+1}, \ldots,  g_n while making progress on lowering the value of the most incorrect constraint, g_i. We leave experiments on this strategy to future work in order to not detract from the focus on breaking these current detection defenses.
>
> > For Honeypot Defence and Sensitivity Inconsistency Detector, what is the impact of the threshold ϕ on the effectiveness of SPGD and OPGD?
>
> This is the detection threshold for the ROC curve. Adjusting the threshold increases or decreases the false positive rate based on the desired sensitivity of your detection model. This information is incorporated into both the Success Rate (SR) numbers and the ROC curve.
>
> > Why is OPGD much less effective than SPGD against Dense Layer Analysis when ϵ=0.01 (Figure 3)?”
>
> We don’t completely know. We hypothesize, however, that because OPGD is "overly-constrained" to never leave the boundary once we reach a valid region, the attack might not be able to find the same solution that SPGD can. Our main argument in this paper for presenting these new attacks is not to say that one attack is the best attack, but to suggest a new general strategy to attack detection defenses that papers might consider in the future. So here it is useful to know that while in most cases OPGD is better, there still do exist cases where SPGD, a relaxation of OPGD, can perform better.
>
> > Why isn’t SPGD evaluated against Steganalysis?
>
> As we say in the paper, “for computational efficiency, we only run our Orthogonal attack as the detection model has a throughput of one image per second.”

---

### Official Review · Reviewer_AXWZ · 2021-11-02

**Correctness:** 3
**Technical Novelty And Significance:** 3
**Empirical Novelty And Significance:** 3
**Recommendation:** 8
**Confidence:** 4

**Details Of Ethics Concerns:**

None.

**Main Review:**

### Pros
Breaking the defenses of adversarial example detection is an important yet underdeveloped field. The approach proposed in this paper is straightforward and effective in practice. The experimental evaluations seem reasonable.

### Cons
The proposed optimization formulation is based on intuition. It would be better if the author can provide theoretical evidences that can explain its effectiveness.

Currently, the empirical evaluations mainly shows its performance against the detection defense. It can be improved by adding more analytical results.





**Summary Of The Paper:**

This paper targets on attacking the defensive mechanism of adversarial examples detection. It proposes a new optimization algorithm to simultaneously meet two different requirements. It verifies its effectiveness on several state-of-the-art adversarial example detection methods.

**Summary Of The Review:**

I would love to recommend an accept for this paper, considering its topic of research, effectiveness, and novelty.

In addition, I want to clarify that I served as a reviewer of this paper for its last submission. I am glad to see that the authors followed my suggestions and made several improvements on their writing. For the last submission, I give a weak accept score in the final stage. Considering that the paper has been improved on several aspects, I raise my score to accept.

---

> ### Author Response · Authors · 2021-11-18
> **Author Response**
>
> We thank the reviewer for asking this question.
>
> While we would love to provide further analytical results, we are not sure what kind of analytical results the reviewer is looking for. If the reviewer could clarify this we can try and provide additional information.
>
> The intent of our paper is to empirically show that the attacks we develop are very effective. This has been common practice for quite some time; we note that (Tramer et al. 2020 “On adaptive attacks” @NeurIPS, Uesato et al. 2018 “Adversarial Risk...” @ICML, Athalye et al. 2018 “Obfuscated gradients” @ICML, and Croce et al. 2020 “Reliable evaluation” @ICML) also do not include any new extensive theory for why their attacks work, other than that it breaks the proposed defenses.

---

### Official Review · Reviewer_Tn7g · 2021-11-05

**Correctness:** 3
**Technical Novelty And Significance:** 3
**Empirical Novelty And Significance:** 2
**Recommendation:** 8
**Confidence:** 3

**Main Review:**

Strengths:

-  The proposed approach is well motivated and novel. Empirically the contributions of the paper are good.
-  Experiments are well designed. The ROC curves help in better understanding.
-  The discussed defences have been reproduced well.

Weaknesses:

-  A more detailed discussion on the past works which broke adversarial-detection defences needs to be done in order to better highlight the novelty of the proposed approach. Could the authors clarify why these detection methods cannot be broken using existing adaptive attacks? [1,2]
-  The clarity on the discussion related to the four defences that are broken could be improved. It is difficult to understand the defence approaches from Section 4 of the paper.
-  Could the authors clarify why an epsilon bound of 0.01 was used to draw the ROC curves? Could the authors share the curves for an epsilon bound of 0.031 (8/255)?

[1] Carlini et al., Adversarial Examples Are Not Easily Detected: Bypassing Ten Detection Methods
[2] Tramer et al., On Adaptive Attacks to Adversarial Example Defenses

##### Update post rebuttal #####

The authors' rebuttal sufficiently addresses my concerns and I am happy to update my recommendation to "Accept".




**Summary Of The Paper:**

The authors propose an attack that could break 4 adversarial detection methods published recently. Traditionally, attacks against detection methods have attempted to maximize the loss for both classification and detection simultaneously. However, using a toy example the authors show that this is suboptimal, as it may not find the worst-case adversaries. The authors propose to minimize the loss (targeted attack setting) iteratively by optimizing either only for the classification pipeline or the detection pipeline at a time.  The attack first considers the classification loss and further tries to fool the detection pipeline until the classification prediction remains incorrect. The authors also propose a variant of the attack by considering gradient steps for the classification pipeline to be orthogonal to the gradients of the detection pipeline and vice versa. Finally, the paper shows that these two proposed attacks completely circumvent four recent adversarial detection methods.

**Summary Of The Review:**

The proposed attack seems interesting. I think the main problem with the paper is with respect to the clarity in writing, especially in Section-4. Further, the past methods which broke detection methods need to be discussed in detail to highlight the significance of the proposed method.

---

> ### Author Response · Authors · 2021-11-18
> **Author Response**
>
> We thank the reviewer for their comments.
>
> > Could the authors clarify why these detection methods cannot be broken using existing adaptive attacks
>
> The reason we did not implement existing adaptive attacks is that the papers we attacked had already tried exactly these forms of adaptive attacks, and showed it was not effective. The Dense Layer Analysis paper, for example, uses exactly the suggested Carlini&Wagner loss function code and loss function: “We use the loss function defined by Carlini and Wagner [14], which the authors use to attack similar detection-based defense methods. Here, we profit from the code the authors published to reproduce their results [5].” (Page 10, section 4.6). However, as the authors show, this existing attack can not defeat their defense. Our improved attack does defeat their defense.
>
> The Trapdoor defense likewise performed this evaluation initially, and also found it was ineffective “Finally, an oracle attacker knows the Trapdoor, and can try to evade detection by jointly minimizing the loss of the adversarial example and the probability that the example is detected. [...] Mathematically, this joint attack loss function is written as [a lagrangian loss function]” (Page 12, https://arxiv.org/pdf/1904.08554v5.pdf). Again, our attack works.
>
> > The clarity on the discussion related to the four defences that are broken could be improved
>
> We attempted to update Section 4 to explain the defense approaches better -- here it would help if the reviewer could clarify what was found most confusing so we can address that specifically.
>
> > Could the authors clarify why an epsilon bound of 0.01 was used?
>
> As we explain in Section 5, we decide to use an epsilon bound of 0.01 because “several of these papers exclusively make claims using the value of 0.01 (SWW+20; SKC+20; TZLD21), but the value 0.031 = 8/255 is more typical in the literature (TCBM20).” This allows us to be fair in our evaluations. Note that because 0.01 is *harder* than 0.031, a break at 0.01 is strong enough to know that it’s also broken at 0.031.

---

### Decision · Program_Chairs · 2022-01-20

**Decision:**

Accept (Poster)

**Comment:**

The authors propose two new variants of (projected) gradient descent for attacking a classifier and a detector simultaneously. Using these two new variants they are able to break four recent detection methods for adversarial samples.

Strength:
- All the reviewer acknowledge that breaking these four defenses is a valuable contribution.

Weakness:
- From a technical perspective the paper is rather simple and no theoretical support for the suggested variants is provided. The justification is rather handwavy. From an optimization perspective it is unclear why not a simple penalty-based approach would have given the same results or would even work better. The choices maded in this paper seem a bit arbitrary and are mainly justified by the fact that they work for the four detectors
- the honeypot defense was already broken in
A Partial Break of the Honeypots Defense to Catch Adversarial Attacks, Nicholas Carlini, arXiv:2009.10975

Minor:
- the authors should update the references, several papers have appeared in the meantime

While I appreciate the contribution of the broken defenses, in terms of technical contribution and discussion of the methods this paper is borderline.